# Complexation of Amino Acids with Cadmium and Their Application for Cadmium-Contaminated Soil Remediation

**Wenbin Yao [1,2], Zhihui Yang [1,2,\*], Lei Huang [3] and Changqing Su [4]**

1 School of Metallurgy and Environment, Central South University, Changsha 410083, China; wenbinyao2020@126.com
2 Chinese National Engineering Research Center for Control and Treatment of Heavy Metal Pollution, Changsha 410083, China
3 Guangzhou University-Linköping University Research Center on Urban Sustainable Development, Guangzhou University, Guangzhou 510006, China; znhuanglei@foxmail.com
4 College of Resources and Environment, Hunan University of Technology and Industry, Changsha 410205, China; changqingsu522@hutb.edu.cn
* Correspondence: yangzh@csu.edu.cn; Tel.: +86-731-88830875

**Abstract:** The interaction of amino acids with toxic heavy metals influences their immobilization and bioavailability in soils. However, the complexation ability of amino acids with Cd has not been well studied. The complexes of amino acids and cadmium were investigated by density functional theory (DFT) calculations and Fourier transform infrared spectrometry (FTIR) analyses. The complex structures were found to be [COc, COc] for fatty amino-cadmium and $PheCd^{2+}$, [COc, COc, COs] for $GluCd2+$ and $ThrCd^{2+}$, respectively. The complex energy of these conformers followed the order $PheCd^{2+}> AlaCd^{2+} > LeuCd^{2+} > GluCd^{2+} > GlyCd^{2+} > ThrCd^{2+}$. Importantly, all of the complex energy values were less than zero, indicating that these complexes could be easily dissolved in water. The $Cd^{2+}$ concentration decreased with increasing amino acid concentration in aqueous solution. The complex stability constants (logβ) followed the order $PheCd^{2+}> AlaCd^{2+} > LeuCd^{2+} > GluCd^{2+} > GlyCd^{2+} > ThrCd^{2+}$, consistent with the order of the calculated complex energy values. The Cd removal efficiencies by Thr, Glu, Gly, Ala, Leu, and Phe were 38.88%, 37.47%, 35.5%, 34.72%, 34.04%, and 31.99%, respectively. In soil batch tests, the total Cd concentration in soil decreased in the presence of amino acids, while the Cd concentration in water increased from 231.97 µg/L to 652.94~793.51 µg/L. The results of sequential extraction showed that the acid-extractable fraction and the reducible fraction of Cd sharply decreased. Consequently, the significant features of amino acids along with their biocompatibility make them potentially applicable chelators in Cd-contaminated soil remediation processes.

**Keywords:** amino acid; cadmium; soil remediation; chelator

## 1. Introduction

"Itai-Itai disease", which occurred in Japan during the 1950s and aroused worldwide concern, was attributed to the prolonged oral intake of cadmium-contaminated rice [1]. Unlike organic contaminants, cadmium cannot be removed via microbial or chemical degradation [2] and poses a threat to human health and ecosystems. Thus, the contamination of soils and sediments by cadmium has been recognized as an urgent global problem [3].

A large number of approaches have been developed to remediate Cd-contaminated soils, such as chemical fixation, chemical leaching, electrokinetics, encapsulation, and biosparging [4]. Among them, ex situ soil washing has been successfully applied for decades due to its high efficiency and economic feasibility. Metal bioavailability mainly depends on the chemical speciation of metals and the activity of free Cd ions in soil solution [5]. The ligands that exist in soils are able to form stable complexes with metal ions and affect the mobility and toxicity of heavy metals. The soil washing method using

organic ligands has been widely used to remove Cd from contaminated soil [6]. In soil washing processes, powerful synthetic Cd chelating agents such as ethylene diaminete traacetic acid (EDTA), and diethylene triamine pentaacetic acid (DTPA) have been widely used. However, residual nonbiodegradable synthetic chelators in soils introduce ecological damage to the soil structure through the aggressive extraction of calcium, causing harm to plants [7]. Thus, the use of natural organic ligands to remedy contaminated soil has attracted increasing attention [8–10].

Recently, plant root exudates, a kind of widely existing natural organic ligand, have had a great impact on the bioavailability and mobility of heavy metals in the soil [11–14]. They display lower extraction efficiency than EDTA and DTPA toward heavy metals; nevertheless, they possess the advantages of a soil-friendly ability and biodegradability [15]. Among these exudates, natural amino acids with functional groups have significant effects on the mobility and bioavailability of heavy metals by forming stable complexes with heavy metals [16–18]. The study of Ghasemi et al. (2013) indicated that amino acids could produce stable complexes with heavy metals and affect the mobility of Cd. Thus, natural amino acids might have great potential for the remediation of Cd-contaminated soil.

According to Jones et al [19], the common amino acids in soil are glutamic acid (Glu), alanine (Ala), glycine (Gly), threonine (Thr), leucine (Leu), and phenylalanine (Phe). Gly has been applied for the extraction of Cu, Pb, and Zn from soils, and it was found to be competitive with synthetic chelators [20]. In the amino groups, the carboxyl group is an electron-donating group capable of complexing metal ions, and the amino group is an amphoteric group capable of donating and obtaining electrons [21]. According to Dudev and Lim, despite the proton affinities of amino acids, the structure is a critical factor in determining the stability of metal-amino acid complexes [22]. For example, the nonzwitterionic structures of lithiated arginine are more stable than the corresponding zwitterionic structures since the heteroatoms in side chains play a critical role in the complexity [23]. Thus, the molecular fragments of Cd-amino acid complexes are worth investigating.

Nevertheless, to the best of our knowledge, the complexation mechanisms of the molecular fragments of amino groups with heavy metals have rarely been illustrated in detail. Herein, six typical amino acids in soil, Gly, Leu, Ala, Glu, Thr, and Phe, were chosen as model organic chelators, and their complexation mechanisms with $Cd^{2+}$ were fully investigated via theoretical calculations and Fourier transform infrared spectrometry (FTIR) tests. Theoretical calculation is a widely used method to probe the affinity of amino acids toward metals and to investigate the structures of the obtained complexes [24]. The FTIR spectra could provide characteristic information on the functional groups from experimental perspectives [25]. The objectives of this research are (1) to demonstrate the general ability of the studied amino acids to extract Cd from contaminated soils and (2) to identify the structure–functional correlations based on the structures of the amino acids. The outcome would help to provide information for the selection and optimization of amino acids for Cd-contaminated soil remediation.

## 2. Experimental and Computational Section

### 2.1. Calculation Methods

All theoretical calculations were conducted by using the Gaussian09 program (Gaussian, Inc., Pittsburgh, PA, USA). The density functional theory (DFT) method was employed to understand the stereochemical structural nature and thermodynamic properties at the B3LYP/def2-TZVP level with an SDD effective core potential, which is suitable for Cd complex calculations [22–27]. To evaluate the accuracy of the proposed calculation method, a few well-known compounds were calculated, and both the calculated bond length and the actual bond length are presented in Table 1. The closed results indicated that the B3LYP/def2-TZVP/SDD method is applicable in this study. Relative energies were determined for the geometries by using single point energy calculations. Several possible structures of the amino acid-Cd complexes were considered starting points.

**Table 1.** Comparison between Computed and Experimental Average Metal-Ligand (M-L) Bond Distances (in Å).

| Molecule | M-L | *RM*-L (calc) | *RM*-L (expt) |
|:---:|:---:|:---:|:---:|
| $[Cd(H_2O)_6]^{2+}$ | Cd-O | 2.26 | 2.27 ($\pm$0.04) |
| $[Cd(NH_3)_6]^{2+}$ | Cd-N | 2.35 | 2.37 ($\pm$0.03) |

*2.2. Experimental Section*

2.2.1. FTIR Detection of the Complexes

Ten milliliters of 0.01 mol/L amino acid solution was mixed with 10 mL of 0.1 mol/L Cd solution at pH 7.0 for 30 min. The mixture was placed at 283 K in a water bath until crystallization appeared. These crystals were dried at room temperature and milled with potassium bromide (KBr), obtaining pellets for FTIR tests. All FTIR spectra of the complexes were determined by FTIR spectroscopy (Thermo Great-50) in the scan region of 400–2000 cm$^{-1}$.

2.2.2. Effect of Amino Concentrations on Complexation

To study the equilibrium constants of the complexation process, certain volumes (from 0 to 10 mL) of 1 mmol/L amino acid and 1 mL of 1 mmol/L cadmium chloride solution were mixed, and then the volume of the solution was adjusted to 100 mL. The pH value was adjusted to 7.0 with 0.1 mol/L HCl and NaOH solution. The cation concentration was adjusted to 0.1 mol/L with a 1 mol/L KCl solution. The equations of the complex process are shown as follows:

$$Cd^{2+}(l) + pL(l) \rightarrow L_pCd \ (l) \tag{1}$$

$$\log \beta = \log \frac{(L_pCd)}{(C_{Cd^{2+}})(L)^p} \tag{2}$$

$$\log \frac{(L_pCd)}{(C_{Cd^{2+}})} = \log \beta + p \log L \tag{3}$$

where $L_pCd$ and $C_{Cd}{}^{2+}$ are the concentrations of the complexes and Cd ions, respectively. $\log \beta$ and $p$ represent the complex stability constant and the number of amino acids with Cd, respectively. $L$ is the concentration of amino acid. When the concentration of amino acids was much greater than that of Cd, the concentration of $L_pCd$ was approximately equal to that of total Cd in solution, and the concentration of amino acids was approximately equal to that of total amino acids. The concentration of $Cd^{2+}$ was measured by a Cd ion meter (Orion 9648BNWP) at 296 K.

2.2.3. Effect of Cation Concentration on Complexation

To investigate the effect of cation concentration on complexation, 100 mL of 0.01 mol/L CdCl$_2$ was mixed with a series of 0.1 mol/L amino acids at pH 7.0 (adjusted by 0.01 mol/L NaOH and HCl). The cation concentration was prepared by KCl solution with concentrations varying from 0.05 to 0.5 mol/L. The $Cd^{2+}$ concentration was detected by a Cd ion meter at 296 K.

2.2.4. Batch Soil Experiments

Cd-contaminated soil was collected from the surface layer (0–20 cm) in an industrial field in Changning city, Hunan Province, China (111°40′54.40″ E, 26°05′72.50″ N). The moisture soil was sampled by using an iron shovel, after which it was air-dried at room temperature and then sieved through a 1-mm nylon sieve prior to use. The basic soil properties are shown in Table 2. A series of amino acid solutions from 0.1 to 1 mmol/L with a pH of 6.5 was mixed with soil at a 10:1 liquid-to-soil ratio. The mixture solution was shaken for 48 h and separated by centrifugation at 8000 r/min for 10 mins. After that, the Cd concentration in the separated solution was tested by inductively coupled plasma

optical emission spectrometer (ICP–OES Agilent 5100). The BCR (a method proposed by the European Community Bureau of Reference) [28] sequential extraction method was employed to analyze the Cd transfer among different fractions in soil treated with amino acids. Cd is mainly distributed in four fractions: acid-extractable fraction, reducible fraction, oxidizable fraction, and residual fraction, which were extracted with four extractants by the following steps: (1) 0.1 mol/L acetic acid (OHAC) was added to the soil to extract exchangeable species and a weak acid soluble fraction for 16 h (HOAC-extractable fraction) at 295 K; (2) the solid in the first step was separated by centrifugation at 3000 r/min for 20 min and then washed by distilled water. After that, 0.5 mol/L hydroxylamine hydrochloride was adjusted to pH 2 with $HNO_3$ to extract the reducible metal species bound to Fe–Mn oxyhydroxides (reducible fraction) in the solid for 16 h at 295 K; (3) the solid of step 2 was obtained by centrifugation at 3000 r/min for 20 min and washed with distilled water. After that, 8.8 mol/L hydrogen peroxide was employed to extract oxidizable metal species bound to organics and sulfides in this solid (oxidizable fraction) for 1 h at 358 K and then shaken with 50 mL $NH_4OAC$ (1 mol/L) for 16 h at 295 K; and (4) the solid of step 3 was obtained by centrifugation at 3000 r/min for 20 min. Finally, the collected solid was immersed in Aqua regia (7.5 mL 6 mol/L HCl and 2.5 mL 14 mol/L $HNO_3$) for 2 h to obtain the residual fraction in the solid. The Cd concentrations in the extractants were analyzed with ICP–OES (Agilent 5100).

**Table 2.** Basic properties of the studied soils collected from Changning.

| Soil Components | Values |
| --- | --- |
| Organic matter | 4.88% |
| Clay (<0.002 mm) | 16.92% |
| Sand (2–0.02 mm) | 68.32% |
| Silt (0.02–0.002 mm) | 14.76% |
| pH | 6.7 |
| CEC (meq $NH_4^+$/100 g soil) | 21.41 |
| Total Cd | 20.33 mg/kg |

## 3. Results and Discussion

### 3.1. The Complexes of Cadmium and Amino Acids

#### 3.1.1. The Complexes of Gly, Leu, and Ala with $Cd^{2+}$

Most amino acids exist as zwitterions in which the N-terminus is protonated and the C-terminus is deprotonated [29]. According to the study of Armentrout, four structures of calculated $PheCs^{2+}$ complexes were considered starting points for geometry and vibrational frequency calculations [30,31]. The structures of metal-Gly complexes calculated in previous works were used as starting points for geometry and vibrational frequency calculations for fatty amino acids [32]. The structures of $ThrCd^{2+}$ complexes were considered according to the study of Bowman and P. B. Armentrout [30,32]. For $GluCd^{2+}$, the structures of $GluBa^{2+}$ and $GluLi^+$ complexes proposed by Jeremy T. O'Brien were considered [33]. Gly, Leu, and Ala were classified as fatty amino acids with similar side chains. The possible isomer structures of Gly-$Cd^{2+}$ complexes after calculation are shown in Figure 1. A series of complex structures were found through calculation: [N] structure, where Cd is bound with the backbone amino nitrogen; [CO] structure, where Cd is bound with the backbone carbonyl; [N, OH] structure, where Cd is bound with the amino nitrogen and hydroxyl group; and [$CO_c$, $CO_c$] structure, where Cd is bound with both the oxygens of the carboxylic acid and ammonia, forming a strong H-bond with the carboxylic group of the protonated Gly.

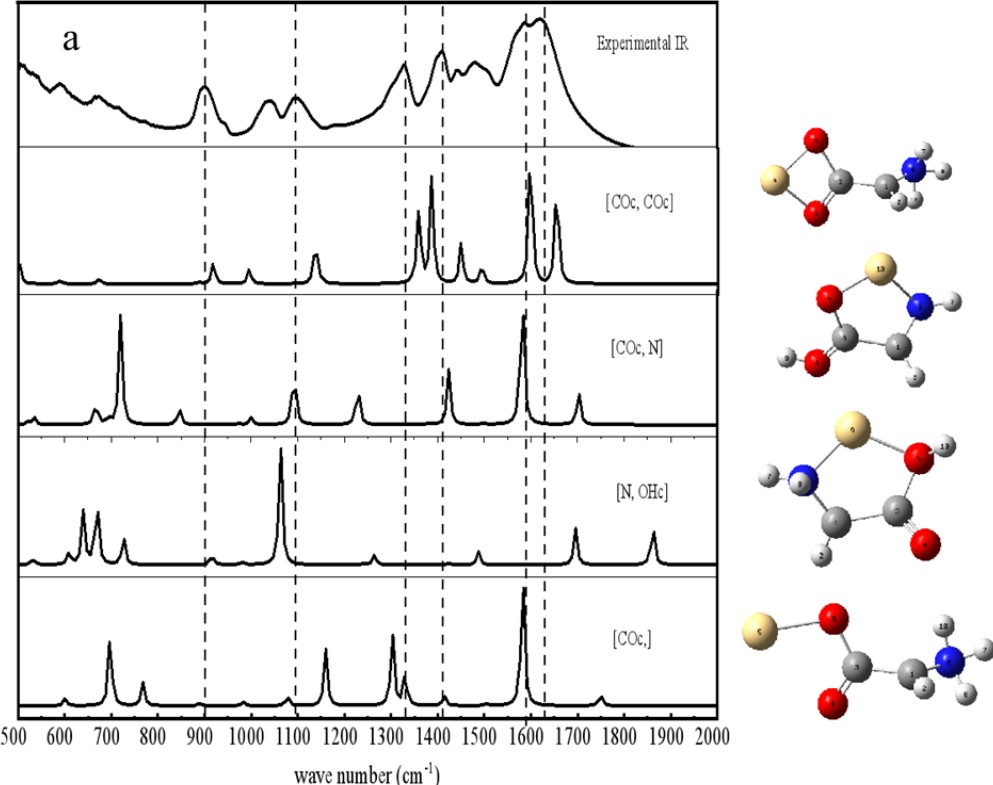

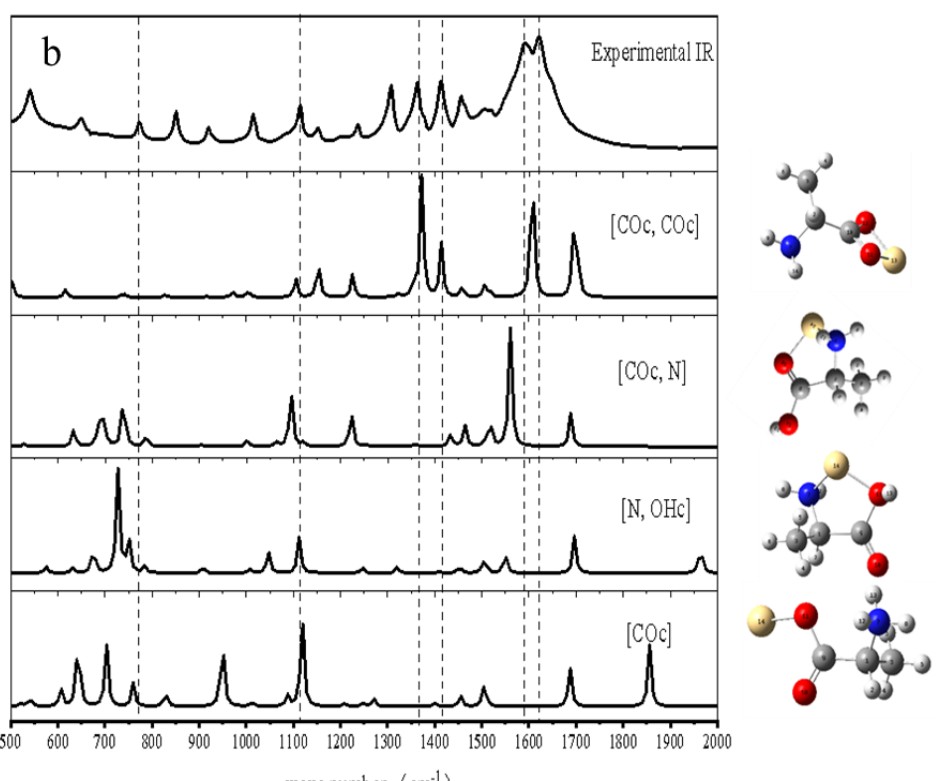

**Figure 1.** *Cont.*

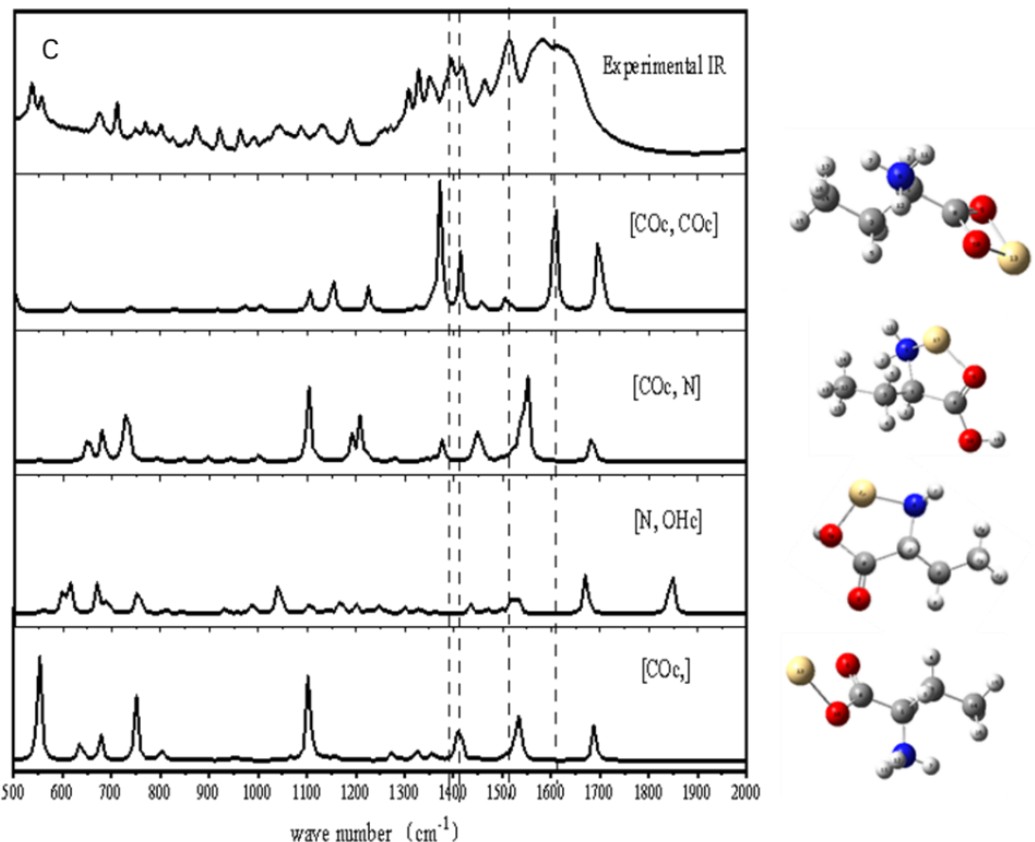

**Figure 1.** The experimental (upper) and calculated (lower) IR spectra of (**a**) GlyCd$^{2+}$, (**b**) Ala Cd$^{2+}$, and (**c**) Leu Cd$^{2+}$. The gray, red, yellow, blue, and silver balls represent carbon, oxygen, cadmium, nitrogen, and hydrogen atoms, respectively.

In the experimental FTIR spectrum of GlyCd$^{2+}$ (Figure 1a), the main characteristic peaks were observed at 1630, 1590, 1480, 1450, 1410, 1325, 1110, 950, and 720 cm$^{-1}$. For the calculated FTIR spectra of GlyCd$^{2+}$, the most intense peaks in the [CO$_c$, CO$_c$] structure of GlyCd$^{2+}$ were relatively well correlated with the experimental spectrum. Briefly, the bands observed at 1640 cm$^{-1}$ (rotation vibration of N-H), 1580 cm$^{-1}$ (bending vibration of N-H), 1500 cm$^{-1}$, 1410 cm$^{-1}$ (stretching vibration of C=O), 1325 cm$^{-1}$ (stretching vibration of C-O), 1100 cm$^{-1}$ (stretching vibration of C-N), and 1040 cm$^{-1}$ were consistent with those of the calculated [CO$_c$, CO$_c$] structure. However, the bands at 1660 cm$^{-1}$ and 1600 cm$^{-1}$ of the [CO$_c$, CO$_c$] structure were redshifted by 20 cm$^{-1}$ compared with the experimental spectrum. Moreover, the bands near 1410 cm$^{-1}$ and 1350 cm$^{-1}$ were redshifted by 25 cm$^{-1}$ and blueshifted by 30 cm$^{-1}$, respectively. The details of complexes are depicted in Table 3. The lengths of Cd-O of [CO$_c$, CO$_c$] were 2.18 and 2.13 Å, which were shorter than those of other structures, and the bond angle of O-Cd-O was 59°. Due to their similar structures, the results of Ala and Leu were similar to those of Gly. As depicted in Figure 1b,c, the isomer structure of AlaCd$^{2+}$ and LeuCd$^{2+}$ was also [CO$_c$, CO$_c$]. As presented in Table 3, the length of the O-Cd bonds was calculated to be 2.23 and 2.28 Å. The O-Cd bond length was longer than that of N-Cd, while a previous study found that the strength of O-Cd was much stronger than that of N-Cd, indicating that the [CO, CO] complex structure was more stable than others [34]. Thus, GlyCd$^{2+}$, AlaCd$^{2+}$, and LeuCd$^{2+}$ mainly exist in the [CO$_c$, CO$_c$] complex structure.

**Table 3.** Calculated bond lengths and bond angles of the amino acid-Cd complexes.

| Complexes | Structure | R Cd-$O_c$ (Å) | R Cd-$X_c$ (N, O) (Å) | R Cd-$Y_s$ (Å) | <$O_c$Cd$X_c$ (°) | <OCd$Y_s$ (°) | <$X_c$Cd$Y_s$ (°) |
|---|---|---|---|---|---|---|---|
| GlyCd$^{2+}$ | [$CO_c$, $CO_c$] | 2.23 | 2.28 | | 59.85 | | |
| | [$CO_c$] | 2.18 | | | | | |
| | [$N_c$, $OH_c$] | 2.23 | 2.16 | | 80.26 | | |
| | [$N_c$, $CO_c$] | 2.28 | 2.16 | | 80.30 | | |
| AlaCd$^{2+}$ | [$CO_c$, $CO_c$] | 2.28 | 2.23 | | 59.93 | | |
| | [$CO_c$] | 2.22 | | | | | |
| | [$N_c$, $OH_c$] | 2.21 | 2.17 | | 79.82 | | |
| | [$N_c$, $CO_c$] | 2.28 | 2.16 | | 80.33 | | |
| LeuCd$^{2+}$ | [$CO_c$, $CO_c$] | 2.23 | 2.28 | | 59.99 | | |
| | [$CO_c$] | 2.18 | | | | | |
| | [$N_c$, $OH_c$] | 2.22 | 2.17 | | 49.87 | | |
| | [$N_c$, $CO_c$] | 2.25 | 2.17 | | 48.60 | | |
| GluCd$^{2+}$ | [$CO_c$, $CO_c$, $CO_s$] | 2.21 | 2.31 | 2.28 | 59.42 | 88.50 | 87.34 |
| | [$CO_c$, $N_c$, $CO_s$] | 2.26 | 2.17 | 2.3 | 115.00 | 98.59 | 76.73 |
| | [$CO_c$, $CO_c$] | 2.30 | 2.27 | | 59.10 | | |
| | [$N_c$, $CO_c$] | 2.23 | 2.17 | | 80.97 | | |
| ThrCd$^{2+}$ | [$CO_c$, $N_c$, $CO_s$] | 2.24 | 2.27 | 2.23 | 76.44 | 85.70 | 75.79 |
| | [$CO_c$, $CO_c$] | 2.23 | 2.27 | | 60.01 | | |
| | [$CO_c$, $CO_c$, $CO_s$] | 2.25 | 2.30 | 2.25 | 60.23 | 72.94 | 71.03 |
| | [N, CO] | 2.45 | 2.27 | | 71.40 | | |
| PheCd$^{2+}$ | [$CO_c$, $CO_c$] | 2.20 | 2.54 | | 48.87 | | |
| | [N, CO] | 2.23 | 2.34 | | 75.09 | | |
| | [N, P] | | 2.20 | 2.33 | 37.53 | | |

*$O_c$: the oxygen atom bound with carbon in the backbone; $X_c$: the nitrogen atom bound with carbon in the backbone or another oxygen atom bound with carbon in the backbone; $Y_s$: the oxygen atom bound with carbon in the side chain.*

3.1.2. The Complexes of Glu with Cd$^{2+}$

Glu has a carboxyl group in the side chain with a dissociation constant (PK$_a$) of 4.25; thus, the dehydrogenation of the side chain and the tridentate structure should be considered [35]. After calculation, several structures were found: [$CO_c$, $N_c$, $CO_s$] structure where oxygen atoms are from each of the carboxylic acid groups and all the N-terminal nitrogen atoms coordinate with the metal ion; [$CO_c$, $CO_c$, $CO_s$] structure (proton transferred from an oxygen atom of the carboxylate group to N), where metal bonds to both of the oxygen atoms from the terminal carbonyl and the side chain; [$CO_c$, $CO_c$] and [N, OH] structures were proposed based on previously reported work [3,6]. The experimental and calculated FTIR spectra of GluCd$^{2+}$ are shown in Figure 2. The most characteristic IR peaks of the [$CO_c$, $CO_c$, $CO_s$] structure were well correlated with those of the experimental FTIR. The characteristic bands of the [$CO_c$, $CO_c$, $CO_s$] structure were observed at 1619 cm$^{-1}$ (stretching vibration of C=O) and 1109 cm$^{-1}$ (bending vibration of C-H), which were in good agreement with the experimental results. The bands at 1690 cm$^{-1}$ (bending vibration of N-H) and 1550 cm$^{-1}$ (vibration of N-H) were redshifted by 15 cm$^{-1}$ and 5 cm$^{-1}$, respectively. The band at 1290 cm$^{-1}$ (vibration of C=O) was blueshifted by 10 cm$^{-1}$. As shown in Table 3, the lengths of O-Cd of the [$CO_c$, $CO_c$, $CO_s$] complex structure were 2.21, 2.31, and 2.28 Å. The lengths of the [$CO_c$, $CO_c$] complex structure were 2.30 and 2.27 Å, which were similar to that of GlyCd$^{2+}$. The O-Cd of [$CO_c$, $CO_c$, $CO_s$] in the terminated carboxy was shorter than that of [$CO_c$, $CO_c$,], indicating a stronger bond between the O atom and Cd$^{2+}$ ion. In addition, according to previous studies, tridentate complexes were

much more stable than bidentate complexes [36–41]. Thus, GluCd$^{2+}$ mainly existed as a stable [COc, COc, COs] complex structure.

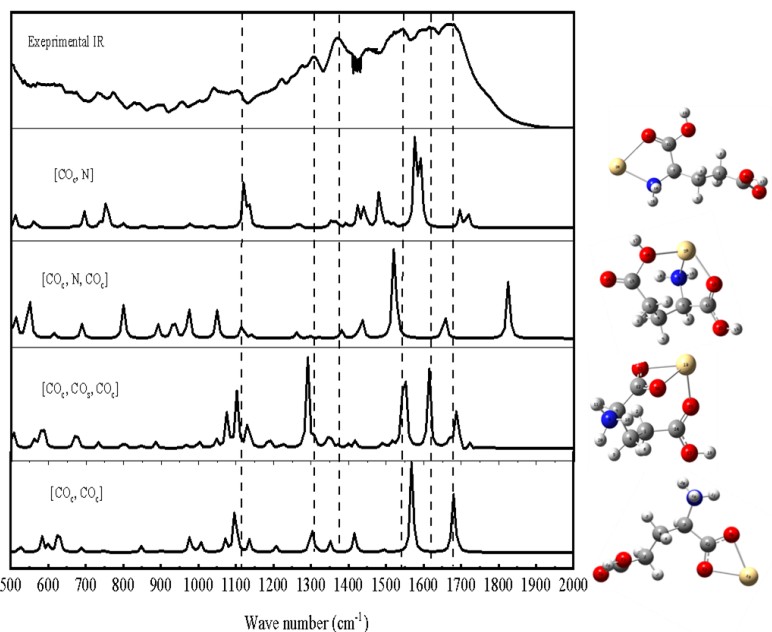

**Figure 2.** The experimental (upper) and calculated (lower) FTIR spectra of GluCd$^{2+}$. The gray, red, yellow, blue, and silver balls represent carbon, oxygen, cadmium, nitrogen, and hydrogen atoms, respectively.

### 3.1.3. The Complexes of Thr with Cd$^{2+}$

ThrCd$^{2+}$ complexes had very similar metal-ligand bond distances for all analogous structures, and the calculations identified that the ground state structure was NZ tridentate [CO$_c$, CO$_s$, N$_c$] for ThrCd$^{2+}$. The experimental and calculated IR spectra of ThrCd$^{2+}$ are shown in Figure 3. It is clearly seen that the calculated spectra of [CO$_c$, CO$_s$, N$_c$] were consistent with the experimental IR spectra. The bands at 1620 cm$^{-1}$, 1460 cm$^{-1}$, and 1340 cm$^{-1}$ were assigned to the stretching vibration of C=O, bending vibration of C-H, and bending vibration of N-H, respectively. All of these calculated bands were well fitted to the characteristic bands of the experimental IR. The bands at 1427 cm$^{-1}$ for the bending vibration of O-H and at 1132 cm$^{-1}$ for the vibration of C-C were redshifted by 16 cm$^{-1}$ cm and 12 cm$^{-1}$ in comparison with the experimental IR. However, the [CO$_c$, CO$_c$] structure was also a reasonable structure for agreement in the lower frequency region. The bands at 1080 cm$^{-1}$ for the C-C rotation vibration exhibited much higher experimental intensities than the theoretically predicted value. Moreover, the peak at 1590 cm$^{-1}$ of the stretching vibration of C=O was blueshifted by 30 cm$^{-1}$ in comparison with the experimental IR. As reported by Wang et al., the amine group of Thr was less protonated at pH values greater than 5.6, and the -NH$_2$ group complexed with Cd, resulting in Cd-N bonds by sharing lone pair electrons [42]. Thus, Cd mainly complexed with NH$_2$ of the amine acids. In Table 3, the lengths of [CO$_c$, CO$_s$, N$_c$] were 2.23 and 2.24 Å, respectively, much shorter than the others. The N-Cd of [CO$_c$, CO$_s$, N$_c$] and [CO$_c$, N$_c$] was 2.27 Å. Thus, the [CO$_c$, CO$_s$, N$_c$] of ThrCd$^{2+}$ was relatively more stable than that of [CO$_c$, N$_c$].

### 3.1.4. The Complexes of Phe with Cd$^{2+}$

PheCd$^{2+}$a naturally aromatic amino acid, possesses a benzene ring with rich electrons; thus, the complexation of Cd with benzene was also considered. As shown in Figure 4, the calculated spectra of [CO$_c$, CO$_c$] were fairly consistent with the experimental IR spectra of the PheCd$^{2+}$ complex. The main characteristic peaks of the actual IR were observed at 1520 and 1330 cm$^{-1}$, which corresponded with the bending vibration of N-H and the stretching

vibration of C-C of the [$CO_c$, $CO_c$] structure. It is clearly seen that both of them fit the actual experimental IR well. Importantly, neither the bands of [$N_c$, P] nor the [$N_c$, $CO_c$] structure could be found in the experimental IR. Moreover, the lengths of the Cd-O/N bonds in [$CO_c$, $CO_c$] (2.20 and 2.54 Å) were much longer than those of the other amino acid-Cd complexes, which was attributed to the attractivity of the benzene ring (seen in Table 3).

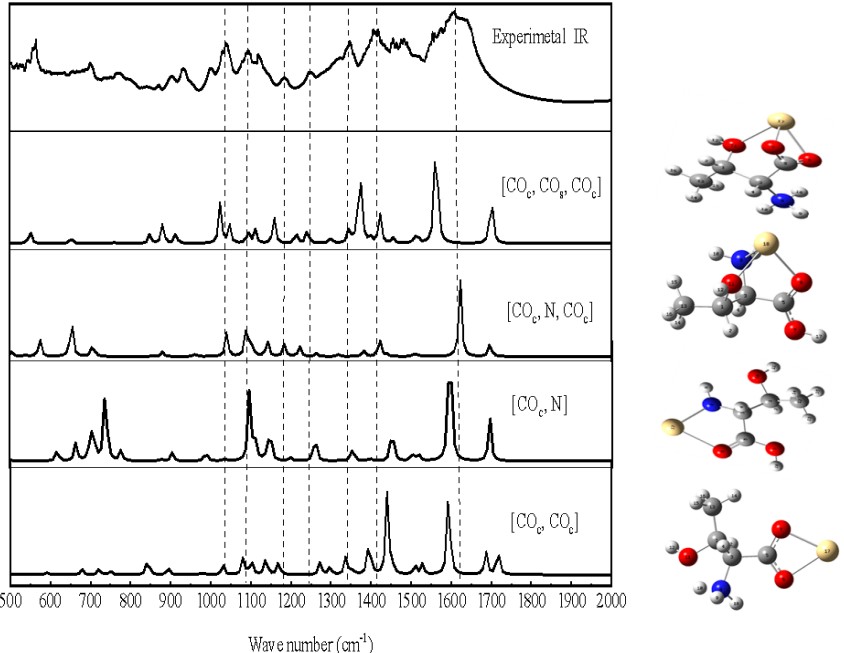

**Figure 3.** The experimental (upper) and calculated (lower) IR spectra of ThrCd$^{2+}$. The gray, red, yellow, blue, and silver balls represent carbon, oxygen, cadmium, nitrogen, and hydrogen atoms, respectively.

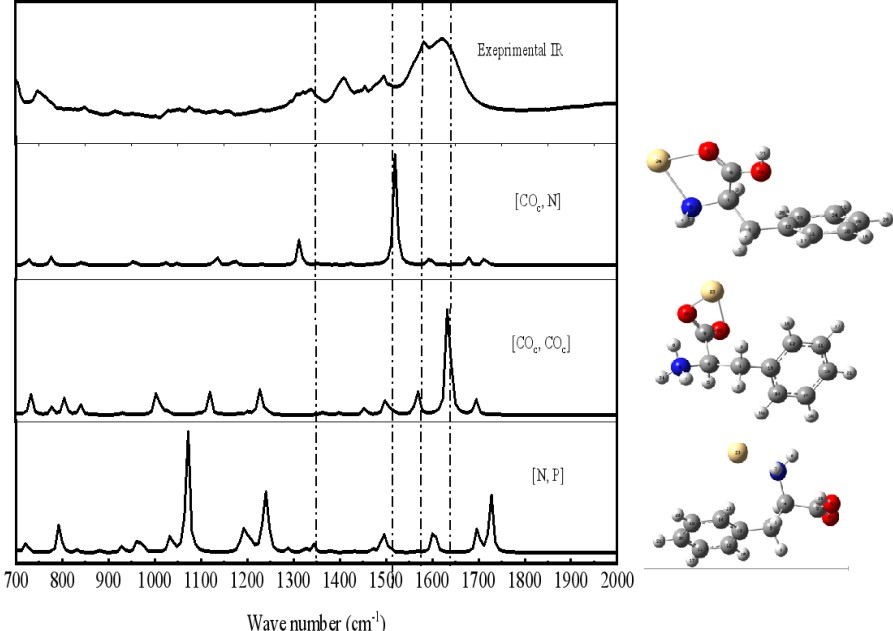

**Figure 4.** The experimental (upper) and calculated (lower) IR spectra of PheCd$^{2+}$. The gray, red, yellow, blue, and silver balls represent carbon, oxygen, cadmium, nitrogen, and hydrogen atoms, respectively.

### 3.1.5. The Theoretical Calculation of the Complex Energy

From the FTIR spectra, the complex structures of Glu and Thr with Cd were found to be tridentate, while those of the others were bidentate. Based on these structures, the free energy of the dissolving processes (Equation (4)) and complex processes (Equation (5)) was calculated in the water phase as follows.

$$LCd^{2+}(s) \rightarrow LCd^{2+}(l) \tag{4}$$

$$Cd^{2+}(l) + L(l) \rightarrow LCd^{2+}(l) \tag{5}$$

As depicted in Table 4, all of the dissolving energies of the complexes were negative, indicating that all these complexes were easily dissolved in water and that all the complex processes in the water phase were spontaneous reactions. The complex energy of Cd and the amino acids followed the order Thr < Glu < Gly < Ala < Leu < Phe. The energy of the Cd and Thr complex process was the lowest, indicating that $ThrCd^{2+}$ was the most stable complex. In addition, the complex energies of $ThrCd^{2+}$ and $GluCd^{2+}$ were −24.33 kcal/mol and −23.87 kcal/mol, respectively, in the water phase, which were much lower than those of the other amino acid complexes with Cd. This might be attributed to the specific steric structures of Thr and Glu. Compared to other amino acids, the additional -OH and -COOH in the side chains of Thr and Glu might donate more electrons to $Cd^{2+}$ [43,44]. Thus, Thr and Glu are better electron donors for Cd ions than other amino acids. In addition, the structures of Gly, Ala, Leu, and Phe were bidentate steric structures, and the energy increased with increasing molecular weight. In the water phase, water molecules inhibit the mobility of the ligands. Thus, the larger the molecule of the amino acid, the more steric energy it will have.

**Table 4.** Calculated energy of amino acids and cadmium complexes.

| Amino Acid | ThrCd$^{2+}$ | GluCd$^{2+}$ | GlyCd$^{2+}$ | AlaCd$^{2+}$ | LeuCd$^{2+}$ | PheCd$^{2+}$ |
|---|---|---|---|---|---|---|
| Complex energy (kcal/mol) | −24.33 | −23.87 | −22.26 | −21.58 | −20.31 | −18.96 |
| Dissolving energy (kcal/mol) | −260.94 | −248.42 | −279.17 | −272.23 | −264.98 | −254.29 |

### 3.2. Pertinent Factors Affecting the Complex of Amino Acids and Cd

The formation of Cd complexes might be influenced by various factors in water. As shown in Figure 5a, the uncomplexed $Cd^{2+}$ concentration increased with increasing ionic strength. In fact, the inhibition might be due to competitive binding onto the active functional groups of amino acids between the cationic ions and $Cd^{2+}$. A similar inhibition phenomenon has been observed in previously reported studies in which cationic ions in natural groundwater negatively affected the complexing of $Cd^{2+}$ with ligands [45]. It is apparent from Figure 5b that the uncomplexed $Cd^{2+}$ concentration decreased with increasing pH, indicating that the complex process was promoted at higher pH. M.T. EI-Halty et al. (1995) also reported that a higher pH was beneficial to ternary and binary chelation formation between metal and amino acids, resulting in more stability than complex formation [46]. This might be because the protonation of amino and carboxyl groups at low pH would reduce the available chelate sites [47]. As reported by Jaime M. Murphy et al. (2019), the proportion of coordinated metal increased with increasing pH, and nearly 100% copper ions were coordinated with amino acids at pH 7.4 [48].

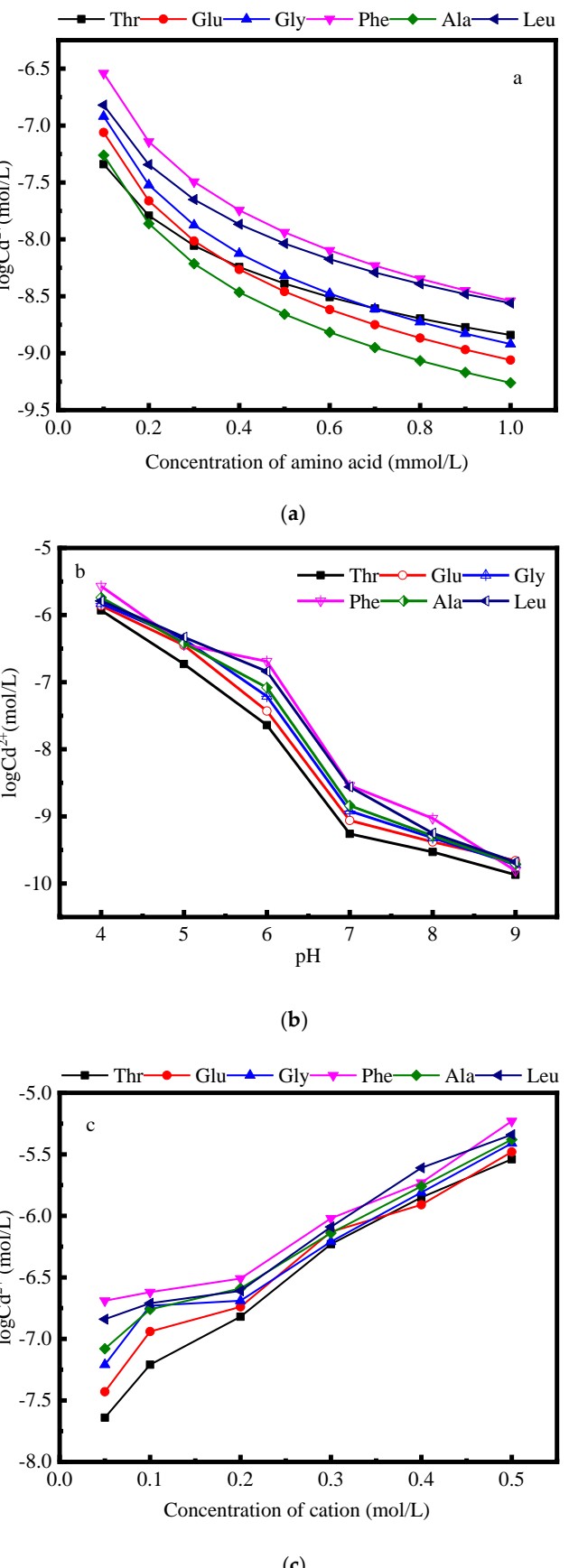

**Figure 5.** The complexation of Cd$^{2+}$ with amino acids at different (**a**) amino acid concentrations, (**b**) pH, and (**c**) cation concentrations.

The complex of amino acids and Cd can also be affected by the initial concentration of amino acids. As seen in Figure 5c, the uncomplexed $Cd^{2+}$ concentration decreased with increasing amino acid concentration from 0 to 0.1 mmol/L. An increase in ligands introduces more soluble complex agents for Cd and decreases $Cd^{2+}$ without precipitation [37]. The complexation of the ligands with $Cd^{2+}$ was fitted, and the fitting curve is shown in Figure 6. The minimum $\log Cd^{2+}$ was estimated to be $-9.26$ mol/L, $-9.06$ mol/L, $-8.92$ mol/L, $-8.84$ mol/L, $-8.56$ mol/L, and $-8.54$ mol/L for Thr, Glu, Gly, Ala, Leu, and Phe, respectively. The complexation constants were then calculated by Equation (3). As presented in Table 5, the resulting constant "$p$" value was less than 1, indicating that cadmium existed in the form of a monodentate complex. Moreover, the constant $\log\beta$ values of these complexes were 5.56, 5.05, 5.03, 4.96, and 4.77, for Thr, Glu, Gly, Leu, Ala, and Phe, respectively. The $\log\beta$ values of these complexes followed the order of Thr > Glu >> Gly > Ala > Leu >> Phe, in good agreement with their corresponding complex energy order in Section 3.1.5. All of these results implied that Thr and Glu had a greater ability to chelate $Cd^{2+}$ to form more stable complexes of $ThrCd^{2+}$ and $GluCd^{2+}$.

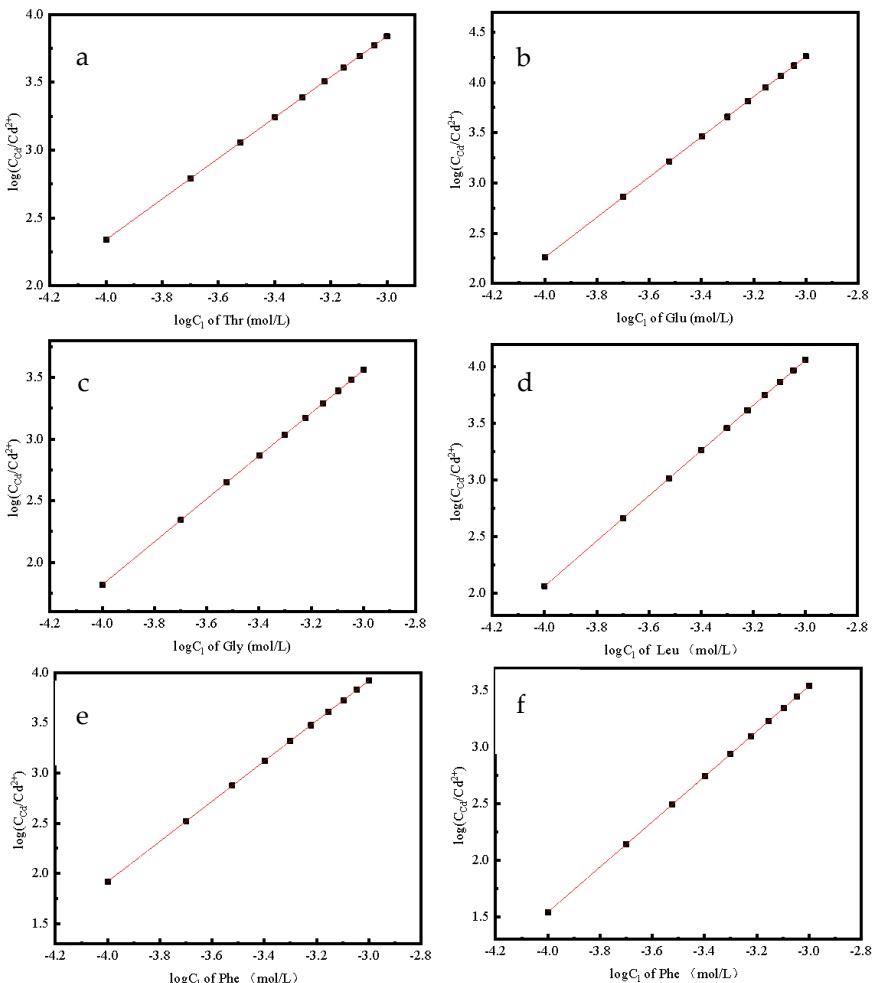

**Figure 6.** Fitted curves of (**a**) Thr, (**b**) Glu, (**c**) Gly, (**d**) Leu, (**e**) Ala, and (**f**) Phe with $Cd^{2+}$.

**Table 5.** Calculated complex parameters of the studied amino acids with $Cd^{2+}$.

| Amino Acids | Thr | Glu | Gly | Ala | Leu | Phe |
|---|---|---|---|---|---|---|
| $p$ | 0.67 | 0.5 | 0.57 | 0.5 | 0.5 | 0.5 |
| $\log\beta$ | 5.56 | 5.13 | 5.05 | 5.03 | 4.96 | 4.77 |
| $R^2$ | 0.99 | 0.99 | 0.99 | 0.99 | 0.99 | 0.99 |

### 3.3. Removal of Cd from Contaminated Soil

Unlike the water phase, the condition in the soil was much more complicated. As shown in Figure 7, the leachable Cd concentration increased with increasing amino acid concentration from 0 to 1000 µmol/L. In the distilled water (control treatment), the leachable Cd was only 231.97 µg/L. The maximum extraction of Cd by the studied amino acids was 793.51 µg/L, 764.75 µg/L, 724.62 µg/L, 708.72 µg/L, 694.66 µg/L, and 652.94 µg/L for 1000 µmol/L Thr, Glu, Gly, Ala, Leu, and Phe, respectively. The Cd removal efficiencies by Thr, Glu, Gly, Ala, Leu, and Phe were 38.88%, 37.47%, 35.5%, 34.72%, 34.04%, and 31.99%, respectively. Thus, it could be concluded that Cd was complexed with the functional groups of amino acids to form soluble complexes, inhibiting the adsorption of Cd on the surface of the soil. Among the six studied amino acids, importantly, Thr and Glu exhibited the highest extraction and removal ability toward Cd in soil, which is consistent with a previous study [7].

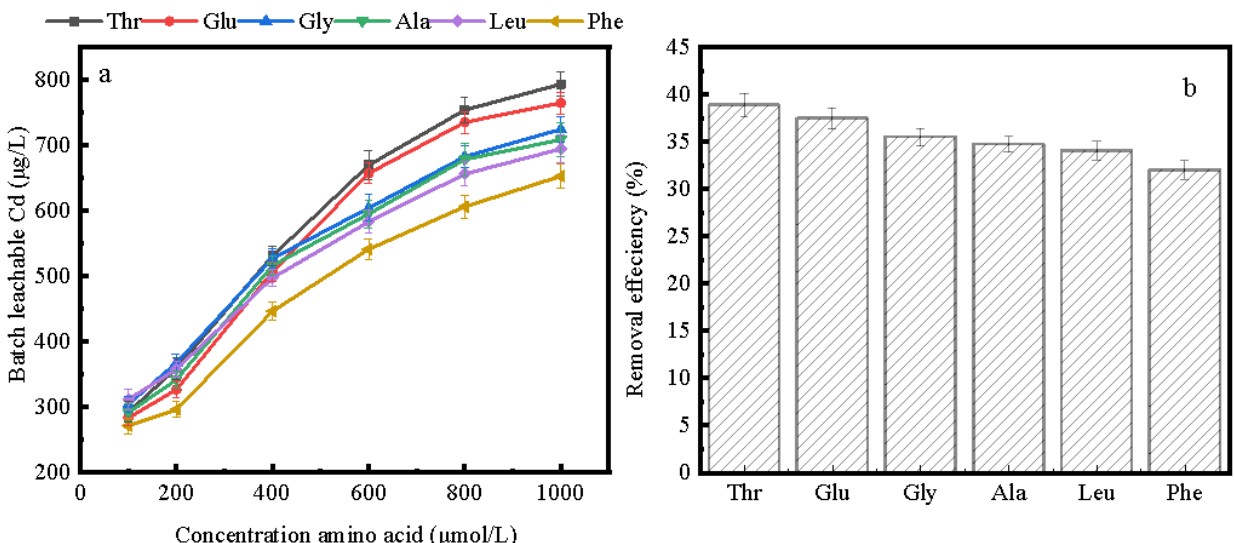

**Figure 7.** (**a**) Cd extraction and (**b**) the removal efficiency of Cd from contaminated soil by using the studied amino acids.

### 3.4. Distribution of Cd in the Soil Fraction

To elucidate the fraction distribution of Cd after treatment with amino acids, BCR sequential extraction was implemented. The acid-extractable fraction and the reducible fraction of Cd with high bioavailability are easily taken up by plants, while the oxidable fraction and the residual fraction of Cd are hardly taken up by plants. In the untreated soil, the distribution of Cd was mainly the acid-extractable fraction [49]. As shown in Figure 8, the acid-extractable fraction, reducible fraction, oxidable fraction, and residual fraction of Cd in the original contaminated soil were 33.81%, 21.36%, 13.73%, and 31.1%, respectively. After amino acid treatment, the acid-extractable fraction and the reducible fraction of Cd decreased dramatically. In the presence of Thr, the acid-extractable fraction of Cd decreased from 33.81% to 16.48%, while the reducible fraction decreased from 21.36% to 14.72%. Similarly, in the presence of Glu, the acid-extractable fraction and the reducible fraction of Cd decreased to 16.58% and 15.24%, respectively. These results meant that the acid-extractable fraction and the reducible fraction of Cd with good mobility in the formation of complexation with amino acids could be easily washed out from soil. Li et al. reported that the acid-extractable fraction of Cd was easily extractable from soil in the presence of complex agents, while the residual fraction of Cd mainly existed in the crystal of minerals and was hardly removed by organic chelating agents [50,51]. The decreasing trend of the acid-extractable fraction and residual fraction of Cd after treatment with amino acids implied that Cd can be easily released into solution to be further taken up by plants or washed out by leachate.

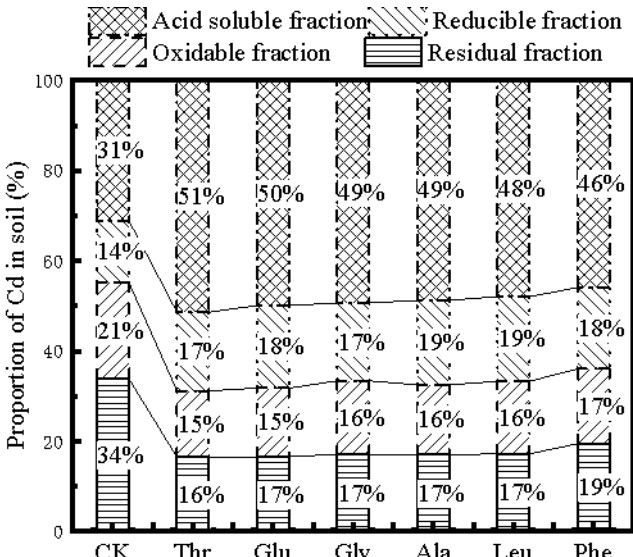

**Figure 8.** The Cd fraction in the contaminated soil using the BCR sequential extraction method.

## 4. Conclusions

In this study, the complexation of amino acids and $Cd^{2+}$ was investigated via theoretical calculations and FTIR measurements, complexation experiments, and soil batch experiments. Importantly, $ThrCd^{2+}$ had the lowest complex energy and the highest logβ value, indicating it was more stable than the others. In the contaminated soil, the removal efficiency of Cd increased with increasing amino acid content. Thr exhibited the best ability to remove Cd from soil due to its strongest affinity toward $Cd^{2+}$, with the lowest complex energy and the highest logβ value among the studied amino acids. The acid-extractable fraction and the reducible fraction of Cd in soil decreased dramatically in the presence of amino acids. Thus, the significant features of amino acids along with their biocompatibility make them potentially applicable chelators in Cd-contaminated soil remediation processes.

**Author Contributions:** Conceptualization, Z.Y.; Formal analysis, W.Y.; Methodology, C.S.; Project administration, Z.Y.; Software, L.H. All authors have read and agreed to the published version of the manuscript.

**Funding:** This research was funded by the National Key R&D Program of China (2020YFC1808002, 2019YFD1100502) and the Project of Science and Technology of Chongzuo City (FA2020008).

**Institutional Review Board Statement:** Not applicable.

**Informed Consent Statement:** Not applicable.

**Data Availability Statement:** Data is contained within the article.

**Acknowledgments:** The authors wish to acknowledge the National Key R&D Program of China (2020YFC1808002, 2019YFD1100502) and the Project of Science and Technology of Chongzuo City (FA2020008) for financial support.

**Conflicts of Interest:** The authors declare no conflict of interest.

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
