# Peer review of "Complexation of Amino Acids with Cadmium and Their Application for Cadmium-Contaminated Soil Remediation"

_applsci, doi:10.3390/app12031114_

Round 1

Reviewer 1 Report

The present study  to demonstrate the general ability of the studied amino acids to extract Cd from contaminated soils, (2) to identify the structure-functional correlations based on the structures of the amino acids, and (3) to provide information for the selection and optimization of amino acids for the Cd contaminated soil remediation. Please check again if all the objectives of the above have been achieved or needing more further studies. 

Methodology- (line 96) All the theoretical calculations were conducted by using Gaussian09 program (Gaussian, 96 Inc.). (Comment: Why do you select  Gaussian09 program? Any literature to support the use of this software? How effective this software can support your hypothesis and your results?

Line 138 -The Cd-contaminated soil was collected from industrial fields in Changning city, Hunan province, China. (Comment: How did you collected the soil samples?? Topsoils? 0-5 cm? Using stainless steel scope?? What are the quality assurance?

Line 142 - After that, Cd concentration in the separated solution was 142
tested by ICP-OES (Agilent 5100).  (Comment: What is the accuracy of the method? Using certified reference materials for soils?? Percentage of recovery??

Line 145- The heavy metals were.. (Comment: You just analysed Cd, why you said heavy metals??

Overall, the science of the whole is acceptable but I suggest to improve the overall flow of readability because the present text is difficult to follow.

Author Response

  1. The present study  to demonstrate the general ability of the studied amino acids to extract Cd from contaminated soils, (2) to identify the structure-functional correlations based on the structures of the amino acids, and (3) to provide information for the selection and optimization of amino acids for the Cd contaminated soil remediation. Please check again if all the objectives of the above have been achieved or needing more further studies. 

Answer: the third objective “(3) to provide information for the selection and optimization of amino acids for the Cd contaminated soil remediation.” was deleted because it is needed for further studies in the future.

  1. Methodology- (line 96) All the theoretical calculations were conducted by using Gaussian09 program (Gaussian, 96 Inc.). (Comment: Why do you select Gaussian09 program? Any literature to support the use of this software? How effective this software can support your hypothesis and your results?

Answer: Gaussian09 program was widely used to analysis the structures of complexes. Hemayat Shekaari et.al (2020) used Gaussian09 program to study the reaction of paracetamol and aminos (https://doi.org/10.1016/j.jct.2020.106348). Sidi Zhu et.al (2020) studied the chelation process of amino trimethylene phosphonic acid with Cu2+ and Pb2+ (https://doi.org/10.1016/j.cej.2019.123711). In this study we calculated the structure of [Cd(H2O)6]2+ and [Cd(NH3)6]2+. The calculated length of Cd-O and Cd-N was 2.26 Å and 2.35 Å. While the experimental length of Cd-O and Cd-N was 2.27 Å and 2.37 Å (the data of experimental length was from reference https://doi.org/10.1021/jp904249s).

  1. Line 138 -The Cd-contaminated soil was collected from industrial fields in Changning city, Hunan province, China. (Comment: How did you collected the soil samples?? Topsoils? 0-5 cm? Using stainless steel scope?? What are the quality assurance?

Answer: The Cd-contaminated soil was collected from the surface layer (0–20 cm) in an industrial fields in Changning city, Hunan province, China (111° 40′ 54.40″ E, 26° 05′ 72.50″ N). The moisture soil was sampled by using an iron shovel, air-dried at room temperature, and then sieved through a 1-mm nylon sieve prior to use.  The above sentences were added in the revised version.

  1. Line 142 - After that, Cd concentration in the separated solution was tested by ICP-OES (Agilent 5100).  (Comment: What is the accuracy of the method? Using certified reference materials for soils?? Percentage of recovery??

Answer: In order to ensure the accuracy of method, the Standard Reference Material of soil and spiked samples were determined. GBW07405(ESS-5) was used as the Standard Reference Material of soil. The measured Cd content in the Reference Soil was 0.456 mg/kg (mean value of three replicates), and the reference value was 0.45mg/kg. The relative standard deviation is 5.59%. In the recovery experiment, the percentage of recovery was 95%.

  1. Line 145- The heavy metals were. (Comment: You just analysed Cd, why you said heavy metals??

Answer: The words “The heavy metals” was replaced by Cd.

  1. Overall, the science of the whole is acceptable but I suggest to improve the overall flow of readability because the present text is difficult to follow.

Answer: We appreciate the reviewer's comment. Yes, we agree with you that there are many language errors in the current manuscript. As the reviewer suggested, we have sent our manuscript to professional company (Springer AJE AI) for the language editing.

Reviewer 2 Report

Manuscript ID: applsci-1541804

Title: Complexation of Amino acids with Cadmium and Their Application for Cadmium Contaminated Soil Remediation

Detail comments

  1. Line 148: the abbreviation of acetic acid should be AcOH.
  2. Experimental, especially batch soil experiments: There is no information as to whether the measurement was performed several times or the experiment was repeated several times.
  3. Line: 302-304: Do the authors consider the presence of cadmium chloride complexes, e.g. [CdCl2] and [CdCl]?
  4. Lines 397-399: It is worth considering this issue carefully, because such “miracles” do not happen.
  5. 5b and 5c: Is the average concentration of Cd ions really in the order of magnitude 10E-11 M?
  6. In the reviewer's opinion, there is no justification for reporting the results of measurements and calculations with more than three significant digits.

Author Response

1. Line 148: the abbreviation of acetic acid should be AcOH.   Answer:The word has been replaced in all article.   2. Experimental, especially batch soil experiments: There is no information as to whether the measurement was performed several times or the experiment was repeated several times. Answer: Three groups are set for each treatment 3. Line: 302-304: Do the authors consider the presence of cadmium chloride complexes, e.g. [CdCl2] and [CdCl]?    Answer: In this article we considered the cation ion strength for the cation could compress the diffusion layer of colloid in water and affect the complex of amino acid with Cd2+(according to https://doi.org/10.1016/j.scitotenv.2021.150668). Thus, the ionic strength was considered. The chloride might also chelator with cadmium, but this complex was easily dissociated. The sentence was replaced by “the uncomplexed Cd2+ concentration increased with increasing ionic strength.”   4. Lines 397-399: It is worth considering this issue carefully, because such “miracles” do not happen. Answer: The conclusion “The acid-extractable fraction and the reducible fraction of Cd in soil decreased dramatically in the presence of amino acids, indicating that Cd was transferred into more stable oxidable and residual fractions.” Was cut off in article.   5. Fig. 5b and 5c: Is the average concentration of Cd ions really in the order of magnitude 10E-11 M? Answer: We used the cadmium ion meter to determine the voltage value (E) of solution, and the calculated the concentration by nest function. The unit of Cd in Fig (5) was wrong, and revised to be mol/L. The standard curve was drawed as fllow:   6. In the reviewer's opinion, there is no justification for reporting the results of measurements and calculations with more than three significant digits.   Answer: All the significant digits was revised to less than three.

Reviewer 3 Report

Please highlight the novelty and objective of the paper in the last paragraph of the Introduction.

Please make sure that this paper is not the "salami-slicing" of the previously paper: 10.21203/rs.3.rs-415354/v1

The abstract and conclusions seem similar. One of them should be modified.

In Materials and Methods, it is necessary to give information about soil physical and chemical properties and location.

Please reduce and summarize the number of paragraphs in the conclusion section.

Could the authors make sure that the English of the written title sounds perfectly correct? 

Author Response

  1. Please highlight the novelty and objective of the paper in the last paragraph of the Introduction.

Answer: The objectives of this research is (1) to demonstrate the general ability of the studied amino acids to extract Cd from contaminated soils, (2) to identify the structure-functional correlations based on the structures of the amino acids. The outcome would help to provide information for the selection and optimization of amino acids for Cd-contaminated soil remediation. The above sentences were added in the revised version.

  1. Please make sure that this paper is not the "salami-slicing" of the previously paper: 10.21203/rs.3.rs-415354/v1

Answer: We appreciate the reviewers for pointing this out. For the purpose of communicating with peers, we published our results in the form of preprints (the salami-slicing one) through Research Square once they became available. This preprint has not been peer reviewed, and it does not affect the publication of our article in official peer reviewed journals. More important, the DOI number of the pre-print version will be combined with the official print version. Moreover, the constructive revision has been made, and the manuscript has been significantly promoted, which is much better than the previously pre-print version.

  1. In Materials and Methods, it is necessary to give information about soil physical and chemical properties and location.

Answer: The Cd-contaminated soil was collected from the surface layer (0–20 cm) in an industrial fields in Changning city, Hunan province, China (111° 40′ 54.40″ E, 26° 05′ 72.50″ N). The soil contained 16.9% of clay, 68.3% of sand and 14.8% of silt. The pH and organic matter content were 6.7 and 4.88%. The above information was added in the revised version.

  1. Please reduce and summarize the number of paragraphs in the conclusion section.

Answer: The conclusion was reduced as suggested.

  1. Could the authors make sure that the English of the written title sounds perfectly correct? 

Answer: We appreciate the reviewer's comment. Yes, we agree with you that there are many language errors in the current manuscript. As the reviewer suggested, we have sent our manuscript to professional company (Springer AJE AI) for the language editing. The detailed language correction has been marked with red color.

Round 2

Reviewer 2 Report

The manuscript in its current form already looks better, but it could and should look much better. May be next time...

Reviewer 3 Report

The manuscript has been carefully revised and addressed well the corresponding questions. Hence, it shall be considered for the publication.